# Assessment of Data-Independent Acquisition Mass Spectrometry (DIA-MS) for the Identification of Single Amino Acid Variants

**DOI:** 10.3390/proteomes12040033

**Published:** 2024-11-06

**Authors:** Ivo Fierro-Monti, Klemens Fröhlich, Christian Schori, Alexander Schmidt

**Affiliations:** 1European Molecular Biology Laboratory’s European Bioinformatics Institute (EMBL-EBI), Hinxton CB10 1SD, Cambridgeshire, UK; 2Faculty of Science, Department Biozentrum, University of Basel, 4056 Basel, Switzerland; klemens.froehlich@unibas.ch (K.F.); christian.schori@unibas.ch (C.S.)

**Keywords:** DIA-MS, single amino acid variants, entrapment database, proteogenomics

## Abstract

Proteogenomics integrates genomic and proteomic data to elucidate cellular processes by identifying variant peptides, including single amino acid variants (SAAVs). In this study, we assessed the capability of data-independent acquisition mass spectrometry (DIA-MS) to identify SAAV peptides in HeLa cells using various search engine pipelines. We developed a customised sequence database (DB) incorporating SAAV sequences from the HeLa genome and conducted searches using DIA-NN, Spectronaut, and Fragpipe-MSFragger. Our evaluation focused on identifying true positive SAAV peptides and false positives through entrapment DBs. This study revealed that DIA-MS provides reproducible and comprehensive coverage of the proteome, identifying a substantial proportion of SAAV peptides. Notably, the DIA-MS searches maintained consistent identification of SAAV peptides despite varying sizes of the entrapment DB. A comparative analysis showed that Fragpipe-MSFragger (FP-DIA) demonstrated the most conservative and effective performance, exhibiting the lowest false discovery match ratio (FDMR). Additionally, integrating DIA and data-dependent acquisition (DDA) MS data search outputs enhanced SAAV peptide identification, with a lower false discovery rate (FDR) observed in DDA searches. The validation using stable isotope dilution and parallel reaction monitoring (SID-PRM) confirmed the SAAV peptides identified by DIA-MS and DDA-MS searches, highlighting the reliability of our approach. Our findings underscore the effectiveness of DIA-MS in proteogenomic workflows for identifying SAAV peptides, offering insights into optimising search engine pipelines and DB construction for accurate proteomics analysis. These methodologies advance the understanding of proteome variability, contributing to cancer research and the identification of novel proteoform therapeutic targets.

## 1. Introduction

Genomic and protein variants can contribute to assessing disease risk and progression [1]. The accumulation of somatic mutations during ageing can lead to cancer and other age-related cardiovascular and neurodegenerative conditions. By far, most genomic variants and mutations are Single Nucleotide Polymorphisms, SNPs [2]. SNPs contribute to complexity and variability in proteomes when propagated into the primary sequence of proteins, such as single amino acid variants (SAAVs), resulting in distinctive proteoforms [3,4] and producing complex and detectable phenotypic changes in organisms. Namely, changes occur in functional, binding or regulatory domains; protein folding and structure; or target amino acids for post-transcriptional modifications or PTMs [5].

Omics-based cancer research has made significant progress, largely attributed to the refinement and increased availability of next-generation sequencing (NGS) methods [6,7]. The widespread adoption of whole genome, exome, and transcriptome sequencing has facilitated the identification of numerous structural and sequence variants implicated in oncogenesis [8]. Despite the comprehensive characterisation at the gene and transcript level, a notable gap exists in understanding the functional impact of these variants on the human proteome.

Proteogenomics, the integration of genomic and proteomics data, provides a comprehensive understanding of cellular processes by identifying variant peptides. Searches for SAAV peptides typically involve searching customised protein DBs compiled from genomic/transcriptomic sequencing data using mass spectrometry (MS) data. This enables the detection of sample-specific sequence variation in proteomics data [9].

In proteomics, searches for variants using MS data can be acquired through two main methods: Data-Dependent Acquisition (DDA) and Data-Independent Acquisition (DIA). DDA involves the sequential fragmentation and analysis of the most abundant ions in a sample. While DDA is versatile and well-established, allowing for the identification of a wide range of peptides and proteins, it suffers from stochastic sample bias, potentially missing low-abundance peptides. DIA, on the other hand, involves systematically fragmenting all ions within predefined mass ranges, offering reproducibility and quantification accuracy compared to DDA, as it provides more comprehensive coverage of the proteome [10].

Various tools have been developed for generating custom DBs in proteogenomic analysis, each differing in their support for different types of variants and DB content [11,12,13,14,15]. Recent efforts have focused on establishing workflows and guidelines for detecting variants, mainly using DDA MS data [16,17,18,19].

In this study, we focused on assessing HeLa DIA-MS data searched with commonly used DIA-MS search engine pipelines as part of a proteogenomic strategy to identify SAAV peptides. HeLa DIA data derived from tryptic digests were searched against customised sequence DBs, including HeLa single amino acid sequences using an entrapment approach. The main objective was to implement a workflow based on determining the number of true positive SAAV peptides identified and the search engine’s pipeline error when using a HeLa DIA data-based proteogenomics strategy.

## 2. Methods

### 2.1. Cell Culture

HeLa cells (https://www.atcc.org/products/ccl-2) (accessed on 30 October 2024) were grown in Dulbecco׳s Modified Eagle׳s Medium (DMEM) with 10% foetal bovine serum and (1×) a penicillin–streptomycin solution incubated at 37 °C and 5% CO_2_ until a confluency of around 70% was reached. Cells were washed three times with PBS and then removed from the culture plate by scraping. The resulting cell suspension was centrifuged at 300 rcf, and the supernatant was discarded.

### 2.2. Sample Preparation

Cell pellets were resuspended in lysis buffer, containing 4% SDS, 100 mM TEAB and 10 mM TCEP. The samples were then heated to 95 °C for 5 min and sonicated using a Bioruptor (10 cycles, each 30 s). Proteins were purified and digested with the SP3 approach [20] using a Freedom Evo 100 liquid handling platform (Tecan Group Ltd., Männedorf, Switzerland). In brief, Speed BeadsTM (#45152105050250 and #65152105050250, GE Healthcare) (Cytek Biosciences, Fremont, CA, USA) were mixed 1:1, rinsed with water, and diluted to the 8 μg/µL stock solution. Samples were adjusted to the final volume of 90 µL, and 10 µL of the beads stock solution was added to them. Proteins were bound to the beads by the addition of 100 µL of 100% acetonitrile to the samples, which were then incubated for 8 min at RT, with gentle agitation (200 rpm), followed by a further incubation for 5 min on a magnetic rack. Supernatants were removed and discarded. The beads were washed twice with 160 µL of 70% (*v*/*v*) ethanol and once with 160 µL of 100% acetonitrile. The samples were placed off the magnetic rack, and 50 µL of digestion mix (10 ng/µL of trypsin in 50 mM triethylammonium bicarbonate) was added to them. Digestion was allowed to proceed for 12 h at 37 °C. After digestion, the samples were placed back on the magnetic rack and incubated for 5 min. Supernatants containing peptides were collected and dried under vacuum.

### 2.3. Mass Spectrometry Analysis

#### 2.3.1. LC-MS Analysis—Eclipse DDA and DIA

Dried peptides were resuspended in 0.1% aqueous formic acid and subjected to LC–MS/MS analysis using an Orbitrap Eclipse Tribrid mass spectrometer fitted with an Ultimate 3000 nano-LC (both Thermo Fisher Scientific, Waltham, MA, USA) and a custom-made column heater set to 60 °C using block randomization. Peptides were resolved using a RP-HPLC column (75 μm × 30 cm) packed in house with C18 resin (ReproSil-Pur C18–AQ, 1.9 μm resin; Dr. Maisch GmbH, Ammerbuch, Germany) at a flow rate of 0.3 μLmin^−1^. The following gradient was used for peptide separation: from 2% B to 12% B over 5 min, then to 30% B over 40 min, followed by 50% B over 15 min, and to 95% B over 2 min, followed by 11 min at 95% B. Buffer A was 0.1% formic acid in water, whereas buffer B was 80% acetonitrile, 0.1% formic acid in water.

DDA mode: the total cycle time was approximately 3 s. Each MS1 scan was followed by high-collision-dissociation (HCD) of the most abundant precursor ions with dynamic exclusion set to 30 s. For MS1, the AGC target was set to 250%, with a fill time of 25 ms and a resolution of 120,000 FWHM (at 200 *m*/*z*). MS2 scans were acquired at a normalised ACG target setting of 100%, maximum accumulation time of 50 ms, and a resolution of 15,000 FWHM (at 200 *m*/*z*). The singly charged ions and the ions with an unassigned charge state were excluded from triggering MS2 events. The normalised collision energy was set to 35%, the mass isolation window was set to 1.4 *m*/*z*.

DIA mode: MS1 scans were acquired in the Orbitrap instrument in centroid mode at a resolution of 60,000 FWHM (at 200 *m*/*z*), with a scan range from 350 to 1200 *m*/*z*, normalised AGC target set to 250%, and maximum ion injection time mode set to 50 ms. MS2 scans were acquired in Orbitrap in centroid mode at a resolution of 15,000 FWHM (at 200 *m*/*z*), a precursor mass range of 400 to 900, a quadrupole isolation window of 12 *m*/*z* with a 1 *m*/*z* window overlap, a defined first mass of 120 *m*/*z*, a normalised AGC target set to 1000%, and a maximum injection time of 22 ms. Peptides were fragmented by HCD with the collision energy set to 33%.

#### 2.3.2. LC-MS Analysis—Exploris 480, PRM

In a first step, parallel reaction-monitoring (PRM) assays [21] were generated from a mixture containing 50 fmol of each proteotypic heavy reference peptide. Peptides were subjected to LC–MS/MS analysis using an Orbitrap Exploris 480 mass spectrometer fitted with a Vanquish Neo (both Thermo Fisher Scientific). Peptides were resolved using the same LC gradient as described above. The mass spectrometer was operated in PRM acquisition mode using the same settings as described for the DDA mode above, with the following modification for target peptide MS2 scans: isolation mass window of 0.4 *m*/*z*, a normalised AGC target of 3000%, a resolution of 120,000 (FWHM), and an injection time of 250 ms. For synthetic isotopically heavy-labelled peptides MS2 scans, an injection time of 50 ms and a resolution of 15,000 (FWHM) were used. All raw files were imported into Skyline (Version 21.2) for manual peptide identification.

### 2.4. Retrieval and Filtering of Single Amino Acid Variants

Genome Screen Mutants tab-separated data of coding point mutations from genome-wide screens (including exome sequencing) and the corresponding normalised Variant Call Format (VCF, containing the 5′ shifted variants and 3′ shifted syntaxes for CDS and genome) file (v.98) were obtained from the Catalogue of Somatic Mutations in Cancer (COSMIC) cell line project (https://cancer.sanger.ac.uk/cosmic) (accessed on 15 January 2024) [22].

The data were filtered for non-silent HeLa variants. The corresponding genes were extracted from the GRCh38 reference genome using the following tools: BSgenome (v. 1.70.2, Pagès H (2024), BSgenome: Software infrastructure for efficient representation of full genomes and their SNPs), GenomicFeatures (v. 1.54.1, Lawrence M, et al.,) [23], (AnnotationHub: Client to access AnnotationHub resources), and Ensembldb (v. 2.26.0, Rainer J, Gatto L, Weichenberger CX (2019); [24]), and Biostrings (v. 2.70.1, Pagès H, Aboyoun P, Gentleman R, DebRoy S (2024); Biostrings: Efficient manipulation of biological strings).

The extracted nucleotide sequences were translated to amino acid (AA) sequences. Fuzzy start codons were fixed by assuming that the correct frame offset leads to a stop codon at the final AA position. A total of 88 sequences containing a regular start codon but premature stop codons were excluded from further processing. Finally, the variant mutations from the filtered VCF file were applied to the GRCh38 reference genome and translated to AA sequence using the previously identified start codon offset from fuzzy start codons. The corresponding 233 canonical HeLa variant sequences were exported as a FASTA file.

### 2.5. In Silico Decoy DB Generation for Entrapment Search

In a first step, the substitution frequency of a certain AA by other AA was calculated based on the extracted HeLa variant sequences and represented in a Sankey diagram using Seqinr (v. 4.2-36, Charif, D. and Lobry, J.R. (2007) Seqin{R} 1.0-2: a contributed package to the {R} project for statistical computing devoted to biological sequences retrieval and analysis. In book: Structural approaches to sequence evolution: Molecules, networks, populations. Publisher: Springer VerlagEditors: U. Bastolla, M. Porto H.E. Roman, and Vendruscolo, M.).

Then, all observable peptides in a standard DIA analysis of HeLa lysate (in house; searched with DIA-NN) were used to generate decoy DBs by introduction of a single AA substitution in each peptide using the probabilities determined in the first step to select the position and substitution of AA. The Sankey diagram was generated to visually prove the similar substitution frequencies in entrapment decoy DBs, as in extracted HeLa variant sequences. The generated decoys were filtered against wildtype peptide and HeLa variant sequences to avoid overlap.

### 2.6. Protein Sequence DBs

The GRCh38 reference genome obtained from COSMIC without the introduced variants from the VCF file was translated to an AA sequence and used as a human reference protein sequence DB. This human reference protein sequence DB (89,244 sequences) was concatenated with the 233 canonical HeLa variant sequences to generate the control customised protein sequence search database (CPS-DB, 89,477 sequences) and compiled as a FASTA file.

Each entrapment search DB was generated by adding different ratios (1:1, 1:10, 1:100, and 1: 400) of variant to decoy variant peptide sequences, by randomly sampling the decoys from the decoy DB. Finally, 4 entrapment DBs of 233, 2.33e3, 2.33e4, and 9.4e4 peptide decoys were exported as FASTA files (the number of sequences present in each entrapment DB and their ratios of decoys vs. total targets are described in Section 3.3.

The *Pyrococcus furiosus* canonical protein sequences (2044 sequences), as specified in [25], along with appended shuffled decoy sequences, were used as an entrapment DB and compiled as a FASTA file.

Every entrapment search DB was concatenated with the control customised protein sequence search DB and compiled as a FASTA file before being used as a search DB.

### 2.7. Database Searching

In all searches, default settings for each search engine pipeline were used for DIA or DDA data analysis. For all search engines, full tryptic specificity was set and 2 missed cleavages were allowed, except for DIA-NN searches, where only 1 missed cleavage was allowed. For all search engines, the carbamidomethylation of cysteine was set as a fixed modification. For all search engines, except for DIA-NN, the oxidation of methionine and the acetylation of protein N-termini were set as variable modifications. By default, DIA-NN considers fixed modifications of cysteine residues by carbamidomethylation and allows for variable modifications such as the oxidation of methionine. For DIA-NN, robust LC, single-pass mode, heuristic protein inference, and no shared spectra were enabled, whereas unrelated runs and MBR were disabled.

### 2.8. DIA Data Analysis Pipelines

Fragpipe-DIA (version 18.0, FP-DIA): DIA raw data were analysed using the MSFragger DIA_SpecLib_Quant workflow. In this workflow, peptides are identified by MSFragger directly from raw DIA data. The pipeline appends to the specified search DB a default decoy DB.

Spectronaut (version 18.4, SN): DIA raw data were analysed using this pipeline. The directDIA workflow from Spectronaut 20 and the Pulsar search engine were used to analyse the data. The pipeline appends to the specified search DB a default decoy DB. A limited access to the Spectronaut software (version 18.4, Biognosys, Schlieren, Switzerland) was due to the commercial licence.

DIANN library free (version 1.8.1, DIA-NN): DIA raw data were analysed using this search engine pipeline. DIA-NN predicts an in silico spectral library from the protein sequence DB and then uses the library to search the DIA data. The pipeline appends a default decoy DB to the specified search DB. This mode of DIA-NN is also known as DIA-NN library-free mode.

### 2.9. DDA Data Analysis Pipelines

MaxQuant (version 2.4.13.0, MQ): only DDA raw data were analysed using this pipeline, with Andromeda as a search engine. Searches were performed against a FASTA file that included the specified sequence search DBs concatenated to the pipeline’s default decoy DB.

Spectromine (version 3.2, SM): only DDA raw data were analysed using this pipeline, with Pulsar as a search engine. Searches were performed against a FASTA file including the specified sequence search DBs concatenated with the pipeline’s default decoy DB. The limited access to Spectronaut software was due to the commercial licence.

Fragpipe-DDA (version 20.0, FP-DDA): only DDA raw data were analysed using this pipeline using the default workflow, where peptides are identified by MSFragger using a basic closed search, with no peptide quantification. Searches were performed against a FASTA file that included the specified sequence search DB concatenated with the pipeline’s default decoy DB.

## 3. Results

### 3.1. Overview of the DIA-MS Evaluation Strategy

HeLa cells, a widely studied human cancer cell line, serve as a key model for proteogenomic analysis [18]. In this study, we used tryptic digests of HeLa cell extracts analysed through a proteogenomic approach, initially with the MS proteomics data obtained using a DIA method on an Orbitrap Exploris instrument.

In a proteogenomic experiment, MS proteomics data are integrated and searched for protein or peptide sequence target variants derived from the genome or transcriptome data using a search engine platform. The search space is either represented by a spectral or a sequence search DB. The search DB construction (i) and the search are key aspects, and the identification of SAAV peptides will depend on the composition of the sequence DB searched and the search engine platform (ii) utilised.

(i) Search DB. In short, HeLa cells SNPs extracted from a COSMIC DB Variant Call Format (VCF) file were propagated to the ENSEMBL human transcriptome and translated into protein sequences. A total of 233 SAAV peptides derived from these variant protein sequences were appended to the human canonical reference proteome. This constitutes our control customised protein sequence search DB (CPS-DB, or wt–human–canonical + SAAV DB). In MS-based proteomics, the search space is a critical factor that determines the accuracy and comprehensiveness of peptide identification. The search space refers to the collection of all possible peptide sequences that can be matched to the MS data. This space is defined by the sequence search DB used in the analysis. For instance, when searching for SAAV sequences, the DB must include both canonical protein sequences and all possible variant sequences. A comprehensive search DB ensures that all potential variants are considered, thereby increasing the likelihood of identifying peptides with single amino acid substitutions. However, expanding the search space also increases computational complexity and the risk of false positives. Therefore, optimising the search DB to balance completeness and specificity is essential for accurate proteomics analysis.

(ii) Search engine pipelines. A search engine pipeline or platform usually performs steps to determine the correct hits from the incorrect ones by comparing the ion masses of a MS/MS spectrum with predicted fragment masses derived from theoretical peptides produced by the in silico digestion of proteins in a DB. Each peptide-to-spectrum match (PSM) is also scored to differentiate between the correct and the incorrect hits. Top-scoring PSMs are likely correct, but low-scoring PSMs are not always incorrect identifications, and these mismatches may be caused by noise in the spectra or by missing peptide sequences in the protein sequence DB [26]. In addition, decoy sequences are added to the ‘target protein DB’ to assist with downstream false discovery rate (FDR) control [27]. The search engine pipeline outputs a list of PSMs, which are used as input to post-processing tools such as PeptideProphet [28] and Percolator [29]. These tools combine the search engine pipeline’s scores and other properties to distinguish between target and decoy peptides, thus improving the sensitivity of peptide and protein identification at a fixed FDR. The differences in score distributions are used to model the optimal combination of features and estimate FDR.

We searched DIA-MS data for SAAV peptides using default settings for current DIA search engine pipelines, such as DIA-NN [30], Spectronaut (SN) [31] with the direct DIA workflow, and Fragpipe-MSFragger with the DIA_SpecLib_Quant (named FP-DIA) [32,33] workflow, as described in Methods and Figure 1.

### 3.2. HeLa SAAV Peptide Identification Using Current DIA Search Engine Pipelines

The results of each search engine pipeline and their cross reference in terms of the number SAAV peptides identifications are shown in Figure 2. These results were obtained by conducting searches against our CPS-DB. A Post-Error Probability (PEP) less than or equal to 0.01, or a probability equal to or greater than 0.99 were applied to all the output peptide data.

In total, we identified 10 different SAAV peptides covering eight different SAAV sites. Comparing the results produced by each search engine using our CPS-DB, we observed a high proportion (70%) of commonly identified SAAV peptides by all search engine pipelines. This high overlap suggested a high identification confidence and low false positive rates for all three search engines employed. However, we believed that comparing the search results from each search engine pipeline (see Figure 2) using a Post-Error Probability (PEP) of less than or equal to 0.01, or a probability greater than or equal to 0.99, the filter might not conclusively demonstrate the search engine’s pipeline performance, as each search engine pipeline implements its own FDR strategy and algorithm to determine a peptide-matching score threshold, in addition to its post-processing tools.

To date, target-decoy DB searches (TDSs) are commonly used to estimate FDR in target DBs [27]. Here, decoy and target DB are of equal size, and thus the number of decoy hits resembles an appropriate estimate of false positive hits expected in target DB matches. While this approach is well established for DDA data searches, it was not compatible with some DIA search engines and could not be used in this study. Notably, a trade-off between keeping the search DB comprehensive to account for all potential SAAV sequences and keeping the custom sequence DB compact prevents a high number of false positives in the search results due to an increasing search space.

As a result, we considered that an entrapment sequence method, as recently suggested in [27,34,35], could serve as a valuable strategy for evaluating the performance of multiple search engines pipelines.

### 3.3. Evaluation of DIA Search Engine Pipeline Performance Using Entrapment DBs

In Vaudel et al. [36], a negligible number of tryptic peptides are shared between the human (approx. 19,587–20,245 target sequences [37]) and hyperthermophilic archaea *Pyrococcus furiosus* (2091 target sequences [36]) proteome, allowing the latter to provide a method for detecting random hits (similar to the decoy DB) by containing both the sample sequences (true positives) and the entrapment sequences (false positive as decoys). Using the entrapment sequence search DB as a standard has proven effective in assessing search engine pipeline performance, as well as in evaluating other steps of the MS data analysis workflow [38].

Based on this method, we evaluated the confidence of DIA-NN, SN, and FP-DIA in searching for SAAV peptides using an entrapment DB as a standard searching approach. The entrapment decoys, as opposed to the common reverse or shuffled sequences, were explicitly generated based on similar substitution frequencies as those in the extracted HeLa set of 233 canonical amino acid variant substitutions. Entrapment DBs containing increasing sets of HeLa single amino acid-substituted decoy sequences (1× decoy DB, 10× decoy DB, 100× decoy DB, and 400× decoy DB) were created and appended to the CPS-DB (including the human reference and the SAAV peptide DB; see Methods section and Figure 3B).

The same search engine pipelines and HeLa DIA data used earlier (Figure 2) were used to search the CPS-DB. This was repeated with subsequently increasing entrapment DB sizes, as well as with a *Pyroccus furiosus* (*Pfu*) entrapment DB appended to the CPS-DB (Figure 3A,B), respectively. Usually, the MS-based proteomics search uses a traditional target–decoy strategy (TDS), where the decoy DB is the same size or smaller than the reference (target) DB using the “target-small decoy search strategy” [39]. As our purpose was to evaluate the DIA search platforms in terms of error, considering true positive SAAV peptides and false positives, we used a search strategy that included an entrapment decoy size different from the traditional decoy size (TDS), including multiple entrapment decoy sizes to test their effect on the DIA platform errors.

We deduced that the number of SAAV peptides identified was maintained relatively stable, regardless of the size of the entrapment DB used (Figure 3A), with variations between 8 to 10 SAAV peptides (Figure 4). Increasing the entrapment search DB sizes elevated the SAAV peptide identification error, determined by the ratio between the number of decoys and the number of true positive SAAV peptide targets identified. This error estimation was designated as false discovery match ratio (FDMR):(1)$$FDMR=\frac{N[{SAA}_{decoys}]}{N_{targets}}$$

To compare the different search engines’ pipelines, their FDMRs within the different entrapment decoy DB outputs were compared, without focusing on the correct decoy size for each instance. Here, FP-DIA < SN < DIA-NN output FDMR obtained by each search engine pipeline, respectively, indicated the FP-DIA search engine pipeline as the most conservative and effective, due to its lower FDMR.

### 3.4. DIA-MS and DDA-MS: CPS-DB and Entrapment DB Searches

An experiment was performed to analyse a HeLa tryptic digest using the DIA mode and subsequently the DDA mode on the same MS instrument (Orbitrap Eclipse). The aim was to compare the similarity of FP-DIA’s SAAV peptide identifications when conducting DDA-MS searches using DDA-MS data. Fragpipe-MSFragger (FP-DDA, with default DDA search workflow, [32]), MaxQuant-Andromeda (MQ, [40]), and SpectroMine-Pulsar (SM) were utilised to search the CPS-DB, as along with a *Pfu* entrapment DB appended to the CPS-DB, respectively. The four search engine pipelines (FP-DIA, MQ, FP-DDA, and SM) collectively identified five shared SAAV peptides, which account for approximately 63% of the total eight, as illustrated in the Venn diagram (Figure 5B). Therefore, combining DIA data with a DDA data search using FP-DIA and the three other DDA search engine pipelines increased the number of true positive SAAV peptides.

Based on the previous DIA search engine pipelines data as a reference, we similarly analysed the HeLa DDA-MS data using DDA search engine pipelines and the entrapment DB approach, as illustrated in Figure 5A. For SAAV peptide searches, we searched the raw DDA data using MQ, FP-DDA, and SM against entrapment-DBs. FDMR was estimated using a comparison of multiple entrapment decoy DB search outputs, as opposed to using the correct search DB decoy size for each search engine pipeline. As the size of the entrapment DB increased, the FDMR increased, but the number of true positive SAAV peptides remained relatively constant. According to our analysis, the DDA HeLa data searches using entrapment sequence DBs identified fewer true positive SAAV peptides, as depicted in the Venn diagram in Figure 5D.

In addition, HeLa DIA and DDA data searches against the CPS-DB concatenated with the standard *Pfu* entrapment DB (Figure 3B) assisted in estimating the FDR for each search engine pipeline search, respectively.
(2)$$FDR=\frac{2[N_{decoys}]}{\left[N_{targets}+N_{decoys}\right]}$$

Except for DIA-NN, similar FDRs were observed when DDA searches were performed on HeLa DDA compared to DIA data (Figure 5C). Here, we did not aim for an accurate FDR estimation, but rather to have a base comparison of the output error of the multiple entrapment search across all search platforms. Studies have shown that using a smaller decoy DB (e.g., 1/8th the size of the target) can achieve nearly identical peptide-spectrum matches (PSMs) at 1% FDR compared to the TDS approach, indicating high accuracy in FDR estimation [38].

### 3.5. Validation of SAAV Peptides Using PRM

To confirm and partially validate the results obtained in previous searches, a Stable Isotope Dilution combined with a Parallel Reaction Monitoring (SID-PRM, [21,41]) experiment was implemented to analyse the previously identified SAAV peptides by either DIA-NN, FP-DIA, or SN, as shown in Figure 6. The fragment ion chromatograms of the light endogenous SAAV peptides and the heavy standards are shown in Figure 6B and Appendix A, respectively. Notably, due to length restrictions during chemical synthesis, the longest peptide “LADFGVLHRNELSGALTGLIR” was not included in the SID-PRM validation, but the corresponding SAAV was still validated by the full-cleaved peptide “NELSGALTGLIR”. Overall, we confirmed eight out of nine SAAV peptides identified by DIA-MS through SID-PRM analysis.

## 4. Discussion

Proteogenomic workflows based on searches against protein or peptide variant sequence DBs represent very resourceful tools used, e.g., to identify the protein-coding genome, including peptides or microproteins derived from non-canonical Open Reading Frames (ncORFs) and the discovery of novel proteoform targets for cancer biology and immunotherapy, serving as the basis for clinical/diagnostic, genetic, and molecular studies of human health and disease.

Various current proteogenomic workflows are in use, but there is no unanimous agreement on the optimal workflow for conducting searches for SAAV peptides. In many studies, the predominant method utilised is DDA-MS. Likely, the common use of DDA-MS is attributed to the specific knowledge of peptide mass available in DDA data, as opposed to the broader window in DIA-MS. This knowledge aids in narrowing down the search space and effectively controls error rates, both being critical aspects of any proteogenomic search. Conversely, no statistical models have been developed thus far that can reliably distinguish between wild-type peptides (e.g., post-translationally modified peptides) and peptide variants using DDA-MS or DIA-MS methodologies.

A common challenge in proteomic analysis is the occurrence of indistinguishable spectra when analysing canonical peptides, making it difficult to differentiate between them. This particularly pronounced issue arises when dealing with, e.g., SAAV peptides and wild-type (wt) post-translational modified peptides that share fragment ions, requiring accurate spectral prediction for differentiation.

Proteogenomics searches are limited by incomplete reference DBs, as proteins cannot be detected if their sequences are missing. SAAVs or splice variants often go unassigned or are incorrectly matched due to absent peptides in DBs. The complexity is heightened with canonical (or non-canonical) SAAV peptides as the search space within the DB expands significantly, resulting in a higher false discovery rate (FDR). Using three- or six-frame translations increases the DB size, complicating searches and increasing false positives while risking false negatives. These large DBs require rigorous validation but still lead to false discoveries or missed proteins. For non-model organisms, combining genome, transcriptome, and proteome data is complex. Then, balancing statistical validation to reduce both false positives and false negatives remains a significant challenge, requiring better experimental and computational methods. Recent strategies addressing this propose critical approaches are as follows: (i) implementing rescoring [42,43] to confirm the accurate identification of peptide-forms and (ii) employing a tailored two-step FDR-controlled error proteomic search methodology to manage this challenge effectively [44,45]. In addition, a key statistical challenge in mass spectrometry proteomics is assessing the search tool’s error control accuracy [35].

Here, our objective was to assess the identification of SAAV peptides from DIA data derived from HeLa cells using a few commonly available search engine platforms. We estimated their performance as the number of correct target-SAAV peptides and their corresponding error rates via search DBs, by similar substitution frequencies in the entrapment decoy DBs, as in the extracted HeLa variant sequences. We deemed decoy DB searches useful for comparing the performance of the different pipelines in identifying SAAV peptides and evaluating how error rates change according to the size of the decoy DB when using a target–decoy strategy.

The number of identified SAAV peptides appears relatively low compared to the 233 HeLa extracted variants. We suggest biological and technical factors. Heterozygosity and allelic imbalance: in diploid cells such as HeLa, some SAAVs may be heterozygous, meaning that only one allele carries the variant, while the other remains wild type. This leads to the wild-type protein being more abundantly expressed, further complicating the detection of the SAAV peptide because the wild-type peptides will dominate the MS signals. Allelic expression: some SAAVs may be derived from SNPs in non-expressed or lowly expressed genes. If the corresponding gene is not actively transcribed or translated in HeLa cells under the experimental conditions, the SAAV peptide will not be detectable. Altered proteolytic efficiency: SAAVs can alter protease cleavage efficiency. If the amino acid substitution is near or at a protease cleavage site (e.g., trypsin recognition sites), it may prevent effective cleavage, leading to inefficient generation of the desired peptides during digestion. This would reduce the number of detectable SAAV peptides in the MS workflow. Peptide length and sequence complexity: SAAVs can result in peptides that are either too short or too long for efficient detection. Short peptides might be poorly resolved or difficult to differentiate from background noise, while very long peptides might fall outside the ideal mass range of typical MS analyses. Interference by PTMs: some SAAV peptides might be post-translationally modified (e.g., phosphorylation, acetylation), which adds further complexity to the peptide’s mass and fragmentation pattern. These modifications could either prevent detection of the SAAV peptide or make it more difficult to confidently identify in MS/MS spectra. Physicochemical properties: the presence of a SAAV can significantly alter a peptide’s hydrophobicity, charge, and isoelectric point. These changes can affect peptide solubility, ionisation efficiency, and chromatographic behaviour during LC-MS, making some SAAV peptides harder to detect or identify.

The error rates estimated for the search engines assessed in our study resemble recently reported data [34]. Noticeably, our entrapment strategy estimated FDR rates were comparable when using the same search algorithms such as FragPipe (MSFragger) and SpectroNaut/SpectroMine (Pulsar). Only for DIA-NN, the estimated FDR was found to be above the 1% threshold set. The positive validation of the SAAV peptides found by SID-PRM and the overall higher number of SAAV identifications indicated that DIA is a well-suited and promising approach for SAAV analysis.

All in all, we are aware of the limitations and challenges of proteogenomics when searching for SAAVs. To overcome these difficulties, several strategies could be employed, including computational and experimental improvements. Instead of relying solely on comprehensive reference DBs, custom DBs can be built using genomic and transcriptomic data specific to the sample. This would involve integrating individual-specific variant data (from whole genome or exome sequencing) with proteomic data, ensuring that SAAVs are included in the search space while keeping the DB size manageable. Targeted strategies could be used, focusing on regions of interest such as known exons or predicted variant-containing regions. This limits the search space while ensuring the inclusion of potentially relevant SAAVs. Combining proteogenomics with other omics data (such as transcriptomics and metabolomics) can provide more context for variant detection, narrowing down the search space, and enhancing confidence in discovered SAAVs. Multi-omics integration also helps filter out potential false positives by cross-validating findings across different data types. De novo peptide sequencing methods, which infer peptide sequences directly from mass spectra, could be used to identify SAAVs without the need for a reference DB. This approach may help identify novel SAAVs missed by traditional DB search methods, although it requires rigorous validation to avoid false positives.

It is worth noting that the data validation methods for proteogenomic experiments remain critical to confirm the identification of SAAV peptides via search engine pipelines, especially SID-PRM in our study, allowing for 88,9% verification of our analysed peptide set (eight out of nine SAAV peptides). There are key advantages of stable isotope-labelled (SIL) peptide standards. Since they behave identically to the endogenous peptides during the MS analysis, they provide an internal standard for comparison, correcting for variations in sample preparation, instrument performance, and ionisation efficiency.

Altogether, our study demonstrates DIA-MS effectiveness for identifying sample-specific SAAV peptides in proteogenomic workflows, elucidating strategies to optimise search engine platforms, and assembling DBs for accurate proteomics analyses. These methodologies enable the comprehensive characterisation of the proteome’s variability and diversity, with the use of DIA facilitating the identification of SAAVs within complex cancer proteomes and the discovery of novel proteoform therapeutic targets.

## Figures and Tables

**Figure 1 proteomes-12-00033-f001:**
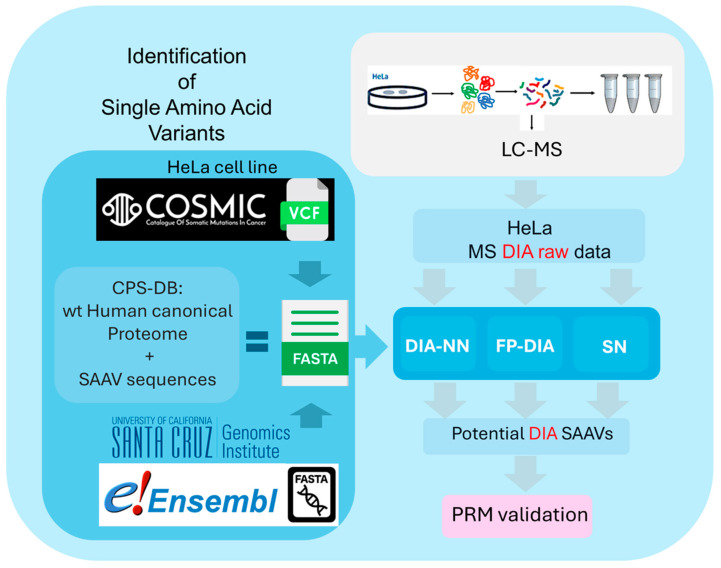
Diagram representing the overall workflow to assess DIA search engine pipelines using a CPS-DB (acronyms for each search engine: Spectronaut (SN), Fragpipe (MSFragger), workflow for DIA (FP-DIA), and DIA–Neural Network (DIA-NN), as described in methods Section 2.8). On the top right corner, HeLa cell tryptic digests are analysed by tandem MS in an MS instrument. Raw HeLa data are analysed by each search engine pipeline to obtain potential SAAV peptides. On the left side, a diagram describing HeLa SNPs extracted from a COSMIC DB Variant Call Format (VCF) file were propagated to the ENSEMBL human transcriptome and translated into protein sequences. These protein sequences, including SAAVs, are appended to the ‘wild type’ human canonical protein sequences reference, resulting in a customised protein sequence search database ‘CPS-DB’ (wt-human + SAAV DB) depicted as a FASTA file. Searches run by each search engine pipeline are against this FASTA file, and the potential SAAV peptides are further validated by parallel reaction monitoring (PRM).

**Figure 2 proteomes-12-00033-f002:**
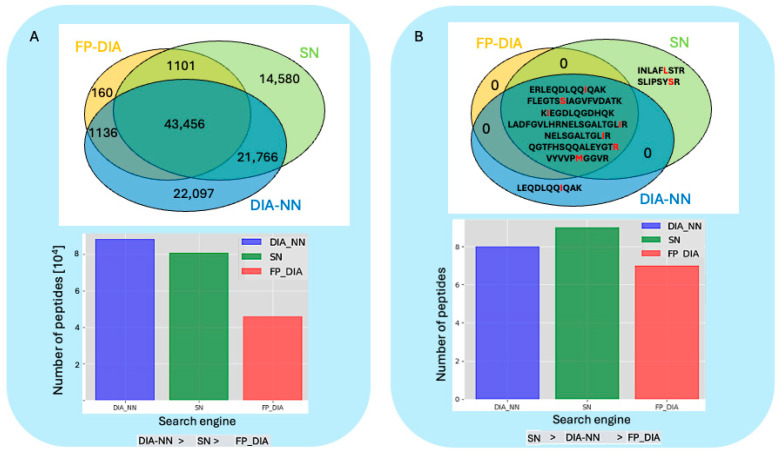
(**A**) HeLa DIA data searches. At the top of the panel, a Venn diagram shows the number of peptides shared among each DIA search engine pipeline output (acronyms for each search engine: Spectronaut (SN), Fragpipe (MSFragger)-workflow for DIA (FP-DIA), and DIA–Neural Network (DIA-NN), as described in Methods, Section 2.8). On the bottom of the panel, a bar plot displays the output of the total number of peptides obtained by each DIA search engine pipeline. (**B**) A bar plot on the bottom of the panel depicts each search engine’s pipeline output number of variant peptides. Below is the list of each variant peptide, indicating the substituted amino acid in red. On the top of the panel, a Venn diagram shows the number of common variant peptides identified by each DIA search engine pipeline.

**Figure 3 proteomes-12-00033-f003:**
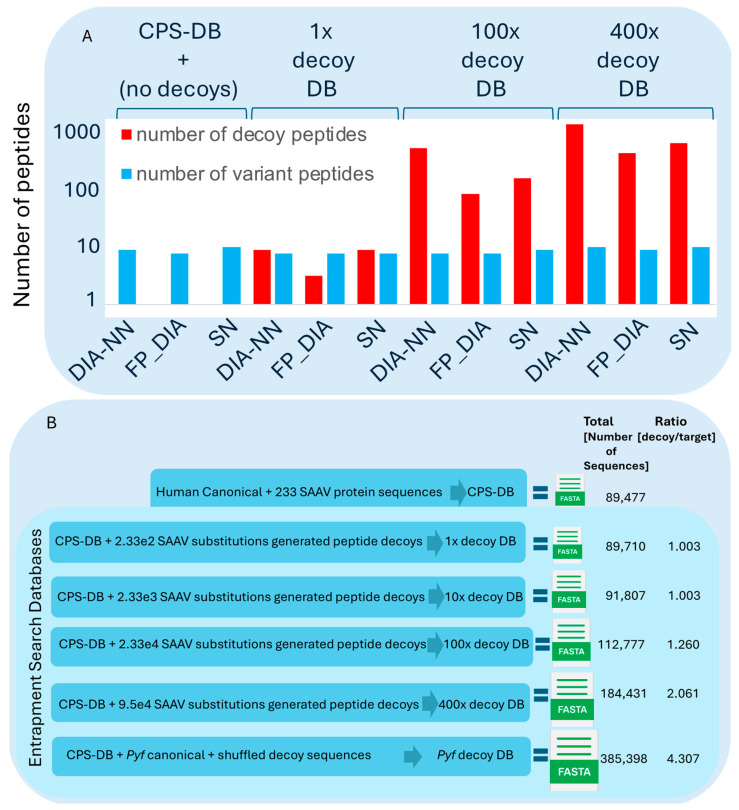
(**A**) Output from individual DIA search engine pipeline searches using the same raw DIA HeLa data and increasing entrapment search DBs size (acronyms for each search engine: Spectronaut (SN), Fragpipe (MSFragger)-workflow for DIA (FP-DIA), and DIA–Neural Network (DIA-NN), as described in methods Section 2.8). The X-axis shows the DIA search engine pipeline used across each search DB (indicated above as the CPS-DB or the entrapment DB included in each search DB). The Y-axis (log10 scale) shows a bar plot with the number of output decoys (red), and the output number of variant peptides (blue). (**B**) Schematic representation of the set of customised sequence search DBs (DB) used in this study, including the CPS-DB, the entrapment DB generated according to the HeLa SAAV substitutions (depicted in increasing FASTA file sizes, not to scale) and the *Pyrococcus furiosus* (*Pfu*) entrapment-DB. On the right side, the total number of sequence hits and the corresponding ratio of number of decoys to the number of targets are shown for each DB.

**Figure 4 proteomes-12-00033-f004:**
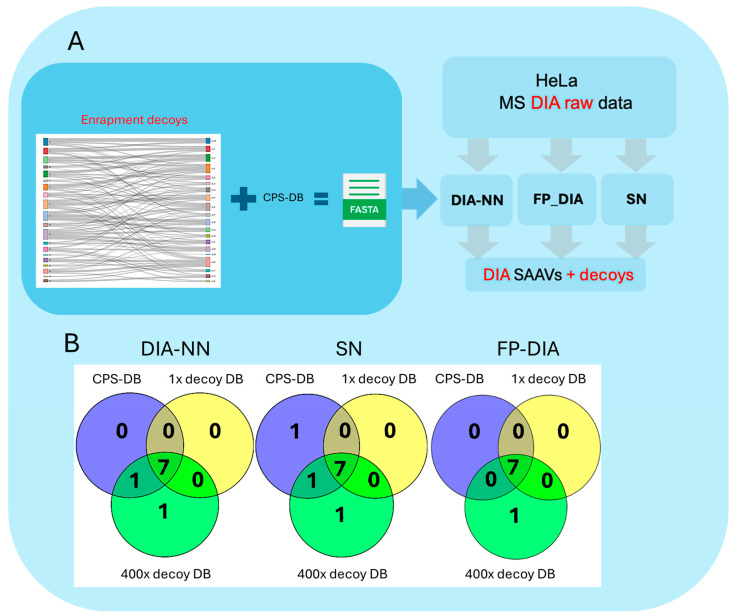
(**A**) Diagram representing the workflow to assess DIA search engine pipelines using entrapment DBs (acronyms for each search engine: Spectronaut (SN), Fragpipe (MSFragger)-workflow for DIA (FP-DIA), and DIA–Neural Network (DIA-NN), as described in Methods, Section 2.8). In the top right corner, the raw HeLa data are analysed by each search engine pipeline to obtain SAAVs and decoy peptides. In the left corner, a Sankey diagram (see Methods section, in silico decoy DB Generation for Entrapment Search) is generated to visually prove the similar substitution frequencies in the entrapment decoy DBs, as observed in the extracted HeLa variant sequences. In addition, the resulting customised protein sequence DB, which includes the human reference, is depicted as a FASTA file appended to each entrapment amino acid substitution sequence peptide DB. (**B**) On the lower-side, Venn diagrams show the number of variant peptides shared between the outputs of each DIA search engine pipeline, filtered for identification scores > 0.99.

**Figure 5 proteomes-12-00033-f005:**
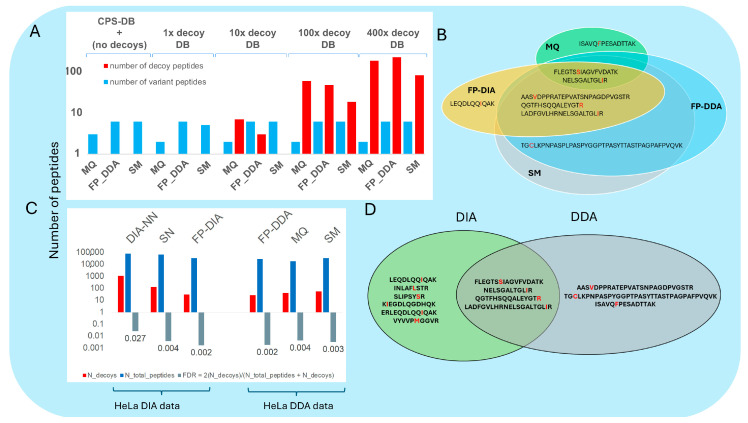
(**A**) Search engine pipeline searches. In the upper left corner, a graph displays the output of individual DDA search engine pipelines searches (acronyms for each search engine: Spectronaut (SN), Spectromine (SM), Fragpipe (MSFragger)-workflow for DDA (FP-DDA), Fragpipe (MSFragger)-workflow for DIA (FP-DIA), DIA–Neural Network (DIA-NN), and MaxQuant (MQ), as described in Methods, Section 2.8 and Section 2.9). The X-axis shows the DDA-search engine pipeline used across each search DB (indicated above as the CPS-DB, along with the number of entrapment decoys or the entrapment search DB included in each search DB search). The Y-axis (log10 scale) shows a bar plot with the number of output decoys (red), and the output number of variant peptides (blue). (**B**) A Venn diagram displays the output variant peptides shared between the DDA search engine pipelines and the output of FP-DIA from the same HeLa sample using the same instrument (Orbitrap-Eclipse). (**C**) A bar plot displaying the output of the DIA and DDA search engine pipelines against the entrapment *Pyf* DB. The upper side shows the DIA and DDA search engine’s pipeline name. The Y-axis (log10 scale) shows the number of detected target and decoy *Pfu* peptides, the total number of target peptides (SAAV + human reference wt peptides), and the corresponding FDR (values shown below each FDR bar), respectively. (**D**) A Venn diagram displays all the output identified variant peptides shared between DIA and DDA pipelines.

**Figure 6 proteomes-12-00033-f006:**
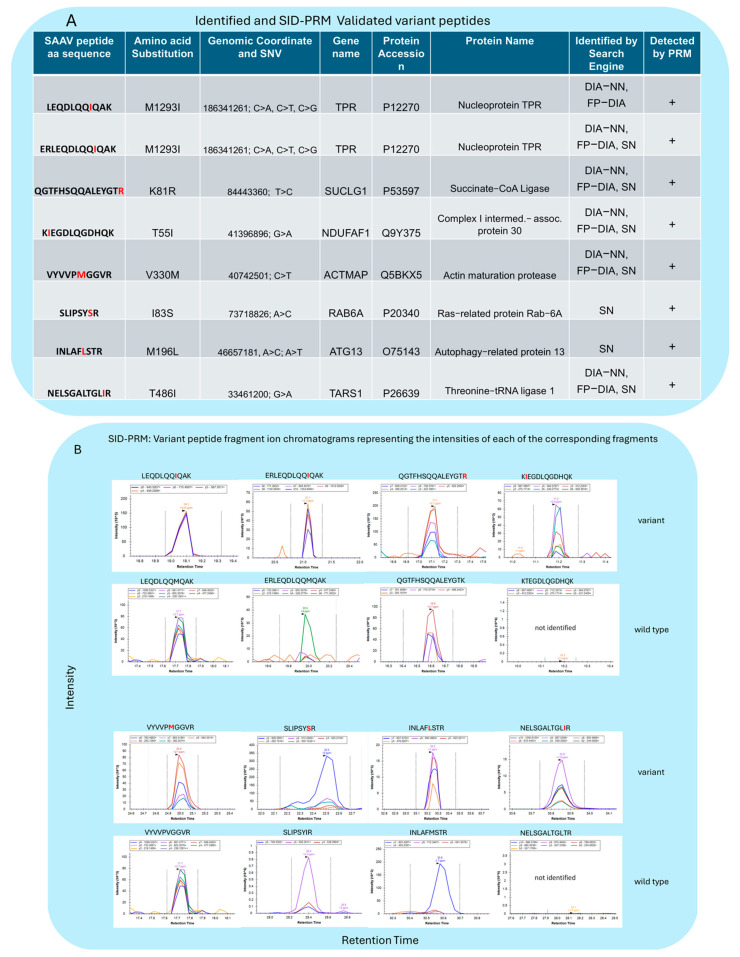
(**A**) A table describes the variant peptides identified (on the left-side) and validated by SID-PRM, including their amino acid sequences and substitutions in red. (**B**) Variant peptide fragment ion chromatograms representing the intensities of each of the corresponding fragments (manually curated based on stable isotope-labelled spike-in peptide standards) versus the retention time.

## Data Availability

The original data presented in this study are openly available in MassIVE at http://MSV000095082@massive.ucsd.edu, with the reference/accession number MSV000095082. A detailed overview of the different datafiles available on MassIVE is also provided in Appendix A.

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
