# Peer review of "Assessment of Data-Independent Acquisition Mass Spectrometry (DIA-MS) for the Identification of Single Amino Acid Variants"

_proteomes, 2024, doi:10.3390/proteomes12040033_

Round 1
Reviewer 1 Report
Comments and Suggestions for Authors
Overall, the manuscript is well-written and presents a solid scientific narrative. However, it lacks a certain degree of novelty and comes across more as a report than a groundbreaking study. There are a few minor issues that can be addressed to enhance the quality of the paper:
1) Section 2.1 describes cell culture growth, but Section 2.2 transitions directly to protein purification. It would be beneficial to include details on how the cell lysate was prepared for protein purification.
2) Line 86, “… with 160 (uL?) of 100% acetonitrile” , missing volume unit
3) Figure 5B, there are only 6 peptides for FP-DIA, while in previously result and discussion, there are 7 peptides for FP-DIA.
4) Does the author count miss cleavage peptide as a different peptide? For example, in figure2, the peptide “LADFGVLHRNELSGALTGLIR” and “NELSGALTGLIR”. So totally there are 9 different peptides from DIA-MS (based on peptide listed on figure 2B). Then in line 405, the author stated “We confirmed 8 out of 9 DIA-MS identified SAAV peptides by SID-PRM analysis…”, the statement is not accurate, because in figure 6A table, the first peptide “LEQDLQQIQAK” does not belong to the peptide list from DIA-MS.
5) Figure 6B, the X- and Y- titles are too small to see clearly.
6) Based on the results, after performing decoy database searches, the total number of peptides and the number of true positive peptides remained unchanged. Additionally, a further validation step using SID-PRM was conducted. Could the authors clarify whether the decoy database searches are still necessary in this context? It would be helpful if they could discuss more on the advantages of using decoy database searches
Author Response
Overall, the manuscript is well-written and presents a solid scientific narrative. However, it lacks a certain degree of novelty and comes across more as a report than a groundbreaking study. There are a few minor issues that can be addressed to enhance the quality of the paper:
Comment 1: Section 2.1 describes cell culture growth, but Section 2.2 transitions directly to protein purification. It would be beneficial to include details on how the cell lysate was prepared for protein purification.
Response 1: We thank the reviewer for pointing this out. We have added the following paragraph to the method section to make the procedure clearer for the readers:
Cells were washed three times with PBS and then removed from the culture plate by scraping. The resulting cell suspension was centrifuged at 300 rcf and the supernatant was discarded. Cell pellets were resuspended in lysis buffer, containing 4% SDS, 100 mM TEAB and 10 mM TCEP. The samples were then heated to 95°C for 5 min and sonicated using a Bioruptor (10 cycles, each 30 sec).
Comment 2: Line 86, “… with 160 (uL?) of 100% acetonitrile” , missing volume unit
Response 2: We thank the reviewer for spotting this. We have corrected this section appropriately.
Comment 3: Figure 5B, there are only 6 peptides for FP-DIA, while in previously result and discussion, there are 7 peptides for FP-DIA.
Response 3: We thank the reviewer for pointing this out. We corrected the figure 5B accordingly (see also next point).
Comment 4: Does the author count miss cleavage peptide as a different peptide? For example, in figure2, the peptide “LADFGVLHRNELSGALTGLIR” and “NELSGALTGLIR”. So totally there are 9 different peptides from DIA-MS (based on peptide listed on figure 2B). Then in line 405, the author stated “We confirmed 8 out of 9 DIA-MS identified SAAV peptides by SID-PRM analysis…”, the statement is not accurate, because in figure 6A table, the first peptide “LEQDLQQIQAK” does not belong to the peptide list from DIA-MS.
Response 4: We thank the reviewer for pointing this out. Technically, miss- and full-cleaved peptides are different peptides and therefore we count them as 2. In this specific case, they cover the same SAAV. While the miss-cleaved peptide does do not add additional information to the results, it confirms the results of the SAAV found with the full-cleaved peptide. We therefore decided to just use the full-cleaved version for targeted MS validation. However, as the miss-cleaved peptide confirms one SAAV and was identified by all three software pipelines, we would still like to include it in the figure. For better understanding, we now mention the total numbers of identified peptides and SAAVs to the manuscript. Counting all peptides identified by DIA results in 10 individual peptides, as one peptide “LEQDLQQIQAK” was only identified by DIA-NN and does therefore belong to the DIA-MS peptide list. To make this clearer, we now show the peptide sequences in the Venn diagram in Figure 2B.
Comment 5: Figure 6B, the X- and Y- titles are too small to see clearly.
Response 5: We thank the reviewer of this comment. We have reordered the figure and show the axis titles in larger fonds.
Comment 6: Based on the results, after performing decoy database searches, the total number of peptides and the number of true positive peptides remained unchanged. Additionally, a further validation step using SID-PRM was conducted. Could the authors clarify whether the decoy database searches are still necessary in this context? It would be helpful if they could discuss more on the advantages of using decoy database searches.
Response 6: We thank the reviewer for pointing this out. We agree that for our samples, the decoy strategy did not have an effect on the number of true positives. It was mainly done to compare the error rates, sensitivity and confidence of the different data analysis workflows applied which was one of the aims of the study. In general, considering the high price of the heavy labeled reference peptides and the additional efforts coming with the setup of targeted MS assays and the MS analysis, a good error rate estimation and low number of false positives is very helpful to keep costs and analytical efforts for this validation step as low as possible. Therefore, having a good error rate estimation in the dataset is still very helpful, also when results are validated. We discussed already the advantages and disadvantages of the different decoy strategies, but have also added one sentence to stress the general benefits of using decoy database searches for this kind of analysis.
Reviewer 2 Report
Comments and Suggestions for Authors
The manuscript is well-written and presents an important contribution to the field. The research is thorough, and the methodology is sound. I believe the manuscript is well-worth publishing. However, I have a few minor suggestions that, if addressed, could further enhance the quality and clarity of the manuscript.
- While the figures provided are informative, I recommend that the data (e.g., in Excel format or similar) should be made available as supplementary materials. This would allow readers and reviewers to examine the data more thoroughly and facilitate a deeper understanding of the results.
- In my opinion, the study would benefit from a discussion of the current challenges associated with single amino acid variant (SAAV) peptide identification.
- I think a critical discussion of the study's limitations is lacking. It may not be clear to the readers, so highlighting the limitations of both the approach and the findings would provide a more balanced perspective and strengthen the manuscript.
After these minor revisions, the manuscript will be well-suited for publication.
Author Response
The manuscript is well-written and presents an important contribution to the field. The research is thorough, and the methodology is sound. I believe the manuscript is well-worth publishing. However, I have a few minor suggestions that, if addressed, could further enhance the quality and clarity of the manuscript.
We thank the reviewer for this positive feedback.
Comment 1: While the figures provided are informative, I recommend that the data (e.g., in Excel format or similar) should be made available as supplementary materials. This would allow readers and reviewers to examine the data more thoroughly and facilitate a deeper understanding of the results.
Response 1: We thank the reviewer for this comment. As stated in the manuscript, all the data associated with the figures, including Excel tables generated by each pipeline search, are, together with the raw data, publicly available via the MassIVE data repository. Most of the proteomics data results files are too big to be provided as supplemental information and are better retrieved directly from MassIVE. However, to make it easier for the reader to find the data, we have now included a detailed overview of the different data files available on MassIVE as well as download links to the corresponding peptide files as supplemental information.
Comment 2: In my opinion, the study would benefit from a discussion of the current challenges associated with single amino acid variant (SAAV) peptide identification.
Response 2: We agree with the reviewer that the manuscript would benefit from a short discussion about the challenges. We have now added a paragraph in the discussion emphasizing the main challenges related to 'search database construction' and 'estimation of the error rate', which are not yet implemented or applied in a standard manner.
Comment 3: I think a critical discussion of the study's limitations is lacking. It may not be clear to the readers, so highlighting the limitations of both the approach and the findings would provide a more balanced perspective and strengthen the manuscript.
Response 3: We agree with the reviewer. We have added a short discussion of the study’s limitations including the efforts of generating SAAV databases, error rate control with the general low number of SAAV peptides over standard, canonical peptides the general low number of SAAV found in this study and the associated need for validation by targeted MS.
Reviewer 3 Report
Comments and Suggestions for Authors
In this work, Fierro-Monti et al. apply Data-Independent Acquistion mass spectrometry to proteogenomics, for the identification of Single-Amino Acid Variants, in order to evaluate DIA as a more sensitive and reproducible acquisition workflow than Data-Dependent Acquisition. A variant database and various decoy databases were queried, and multiple DIA and DDA workflows were benchmarked using HeLa cells, with varying decoy database designs.
Taken together, this is a clear technical piece that could serve as a starting place and template for setting up a proteogenomic DIA workflow, which is increasingly being applied within the proteogenomics field.
However, given that only one HeLa dataset was used, the total number of SAAVs identified is quite small (9 peptides, 8 confirmed by PRM, with 233 SAAVs present in the database), which provides a minimal sample size for comparing results among the various workflows. To this point, DDA methods were able to identify 7 SAAV peptides, which is a small difference which may or may not grow as the approach was applied across multiple systems to capture a larger pool of SAAVs. In order to increase the significance and impact of the findings, and strengthen the case of applying DIA over DDA, I recommend extending the analysis to an additional dataset, such as a high mutation rate cancer cell model with a higher number of SAAVs, or another example of proteogenomics.
Figures are generally good quality and clear, however, the use of Venn diagrams to describe such small absolute numbers is unnecessary (Figures 2B, 4B) and should be changed to match the style of Figure 5C, which lists the individual peptides identified in each case.
In Figure 6 B, it would be more informative and convincing if heavy and light PRM chromatograms are shown side-by-side.
Author Response
In this work, Fierro-Monti et al. apply Data-Independent Acquisition mass spectrometry to proteogenomics, for the identification of Single-Amino Acid Variants, in order to evaluate DIA as a more sensitive and reproducible acquisition workflow than Data-Dependent Acquisition. A variant database and various decoy databases were queried, and multiple DIA and DDA workflows were benchmarked using HeLa cells, with varying decoy database designs.
Taken together, this is a clear technical piece that could serve as a starting place and template for setting up a proteogenomic DIA workflow, which is increasingly being applied within the proteogenomics field.
We thank the reviewer for this positive feedback.
Comment 1: However, given that only one HeLa dataset was used, the total number of SAAVs identified is quite small (9 peptides, 8 confirmed by PRM, with 233 SAAVs present in the database), which provides a minimal sample size for comparing results among the various workflows. To this point, DDA methods were able to identify 7 SAAV peptides, which is a small difference which may or may not grow as the approach was applied across multiple systems to capture a larger pool of SAAVs. In order to increase the significance and impact of the findings, and strengthen the case of applying DIA over DDA, I recommend extending the analysis to an additional dataset, such as a high mutation rate cancer cell model with a higher number of SAAVs, or another example of proteogenomics.
Response 1: We agree with the reviewer that an additional and larger dataset would further strengthen our findings. However, given the fact that we not only take the few SAAVs found for the evaluations, but also base this on all identified canonical peptides and on the fact that SAAVs are also normal, not modified peptides, we think we do have enough datapoints for calculating correct error rates and compare DDA and DIA in a meaningful manner. Nonetheless, we have added this point of low number of SAAVs to the manuscript and discuss potential larger applications for future further validation studies together with the strengths and weaknesses of the approach suggested by Reviewer 2, comment 3.
Comment 2: Figures are generally good quality and clear, however, the use of Venn diagrams to describe such small absolute numbers is unnecessary (Figures 2B, 4B) and should be changed to match the style of Figure 5C, which lists the individual peptides identified in each case.
Response 2: We agree with the reviewer and have changed 2B the match the style of 5B (we guess this was the figure the reviewer was having in mind instead of 5C). For space reasons, we would like to keep the numbers in Figure 4B. As Figure 5B has the information from several experiments, we decided to additionally provide a Venn diagram showing only DIA and DDA results for a clear and easy comparison of DIA and DDA results (see also Reviewer 1, comment 4).
Comment 3: In Figure 6 B, it would be more informative and convincing if heavy and light PRM chromatograms are shown side-by-side.
Response 3: We agree with the reviewer and have now added the light and heavy PRM transitions for all 8 SAAV peptides validated by SID-PRM in supplementary figure S1.
Reviewer 4 Report
Comments and Suggestions for Authors
In the paper named “Assessment of data-independent acquisition mass spectrometry 2 (DIA-MS) for the identification of single amino acid variants” author make and interesting study that give information about the reproducibility and proteome sequence coverage of DIA-MS method. Moreover author describe a new workflow based on determining the number of true positive SAAV peptides identified and the search engine´s pipeline error using a HeLa DIA data-based proteogenomics strategy.
Only minor questions are required
1) In SP3 sample preparation nothing is said about protein reduction and alkylation. Have author made these procedures in protein digestion or only protein digestion on beads?
2) In point 2.3 none about the total gradient time is said? It seems for the information in this point that the gradient is around 90 minutes. Please include this information. Is used the same gradient for DDA and DIA? In point 2.3.2 this data is missing
3) Please include in point 2.7 the default settings used in the search
4) In DIANN analysis the parameters used are missing please include this data.
5) The information about protein number and uniprot database release used in the search are missing
6) In figure 2 legend it will be better to include the DIA-NN, SN and FP-DIA significate in order to facilitate the figure compression.
7) When author say that made a search engine using a CPS-DB (customised protein sequence search database) to better understand the effectiveness of their method the number of proteins and peptides (233 canonical HeLa variant sequences plus the uniprot protein database) should be cited.
8) The paragraph between line 336 and 340 is confusing. They say that only 8 to 10 SAAV were found used entrapment DB (figure 3A) however in figure with 400x decoy database the number is higher, and author say that increasing the search database size the SAAV peptide identification error increase. Please clarify this point
9) The point of peptide validation is very interesting. Have author compare the data obtained using this SAAV with the peptide found in the database? This mean all this peptides has an aa variation shown in Red how about the non aa variation peptide, are both found or only the aa variation is found.
Author Response
In the paper named “Assessment of data-independent acquisition mass spectrometry 2 (DIA-MS) for the identification of single amino acid variants” author make and interesting study that give information about the reproducibility and proteome sequence coverage of DIA-MS method. Moreover author describe a new workflow based on determining the number of true positive SAAV peptides identified and the search engine´s pipeline error using a HeLa DIA data-based proteogenomics strategy.
Only minor questions are required
We thank the review for this positive feedback.
Comment 1: In SP3 sample preparation nothing is said about protein reduction and alkylation. Have author made these procedures in protein digestion or only protein digestion on beads?
Response 1: We thank the reviewer for this observation. We have addressed this issue. Please see reviewer #1 comment #1.
Comment 2: In point 2.3 none about the total gradient time is said? It seems for the information in this point that the gradient is around 90 minutes. Please include this information. Is used the same gradient for DDA and DIA? In point 2.3.2 this data is missing
Response 2: We thank the reviewer for this comment. As the gradient is not linear over the whole-time range, we would refrain from using a simplification such as “a 60 min gradient was used” as this might be misleading. We prefer to provide the reader with a more detailed gradient overview.
Section 2.3.2 states:
“Peptides were resolved using the same LC gradient as described above.”
Comment 3: Please include in point 2.7 the default settings used in the search
Response 3: For all search engines, full tryptic specificity was set, 2 missed cleavages were allowed, except for DIA-NN searches where only 1 missed cleavage was allowed. For all search engines, carbamidomethylation of cysteine was set as fixed modification. For all search engines except for DIA-NN, oxidation of methionine and acetylation of protein N-termini were set as variable modifications. For DIA-NN, robust LC, single-pass mode, heuristic protein inference and no shared spectra were enabled and unrelated runs and MBR were disabled.
Comment 4: In DIANN analysis the parameters used are missing please include this data.
Response 4: We have now added all relevant DIANN parameters (see comment above).
Comment 5: The information about protein number and uniprot database release used in the search are missing
Response 5: We used the protein sequences without the variants are described under 2.6. Protein Sequence DBs: (lines 166-169) "GRCh38 reference genome obtained from COSMIC without the introduced variants from the VCF file was translated to AA sequence and used as a human reference protein sequence DB”. We have now added the corresponding protein number. We have also added the reference of the Pyrococcus furiosus database used.
Comment 6: In figure 2 legend it will be better to include the DIA-NN, SN and FP-DIA significate in order to facilitate the figure compression.
Response 6: We agree with the reviewer and have now added the abbreviations (SN, FP) to the figure legend and reference the method section 2.8 for details.
Comment 7: When author say that made a search engine using a CPS-DB (customised protein sequence search database) to better understand the effectiveness of their method the number of proteins and peptides (233 canonical HeLa variant sequences plus the uniprot protein database) should be cited.
Response 7: We agree with the reviewer that this might be a bit unclear. We have now changed “hits” to “sequences” in Figure 3B and also added the number of sequences to the method section (see also comment 5).
Comment 8: The paragraph between line 336 and 340 is confusing. They say that only 8 to 10 SAAV were found used entrapment DB (figure 3A) however in figure with 400x decoy database the number is higher, and author say that increasing the search database size the SAAV peptide identification error increase. Please clarify this point
Response 8: The aim of this part was to compare the error rates of the different search engines evaluated. In figure 3A the 'number of SAAV peptides' identified (bars in blue) is relatively constant (8 to 10) regardless of the increasing size of the entrapment DB, also for 400x decoy db. This confirms the robustness of our approach. In the same figure 3A the 'number of decoy peptides' identified (bars in red) increases along the increasing size of the entrapment DB, which is expected. This is inferred as an increase in 'identification error', which is defined as: FMDR = N decoys / N targets. We used this FMDR to compare the different DIA analysis tools for each decoy db and here the results are very consistent showing that FP_DIA always had the lowest number of decoy hits while identifying similar number of SAAV target peptides. We have now changed the corresponding paragraph to make this point clearer.
Comment 9: The point of peptide validation is very interesting. Have author compare the data obtained using this SAAV with the peptide found in the database? This mean all this peptides has an aa variation shown in Red how about the non aa variation peptide, are both found or only the aa variation is found.
Response 9: We thank the reviewer for pointing this out. We did indeed find the corresponding peptides without SAAV in a few cases, in others not. To not confuse the reader and keep the manuscript streamlined, we would rather not show these peptides in the manuscript. For readers that are interested in these, they can be obtained from the publicly available data at the MassIVE repository.
Round 2
Reviewer 3 Report
Comments and Suggestions for Authors
I appreciate the authors revisions - the quality of the figures and the clarity of the results has improved.
However, the authors have not addressed what I feel is the primary drawback in the study - the low number of identified SAAVs and the minor difference in coverage of these SAAVs between DIA and DDA methods. While I don't disagree with the authors' point that the number of canonical peptides is sufficient for FDR estimation, my concern was regarding the overall significance of the study justifying publication and supporting the assertion that DIA is performing considerably better than DDA for this purpose, as implied in the below lines.
Lines 497-498 "The overall number of SAAV peptides identified by DIA-MS search engine pipelines ranges from around 6 to 9 and is thus considerably higher than DDA (3-6 peptides)"
The authors have failed to convince me that 9 peptides with DIA is considerably higher than 6 peptides with DDA. To be suitable for publication, as I described in my first review, I feel like an additional study in another cell line would be necessary to support the conclusions in this manuscript regarding the ability for DIA to robustly outperform DDA in identifying SAAVs.
Author Response
We thank the reviewer for pointing this out again. As already stated in the first round of revisions, we agree with the reviewer that the SAAV numbers are rather small for a profound comparison of DDA and DIA-MS. The main aim of the study is to demonstrate that, besides DDA-MS, DIA-MS is also well suited for the identification of SAAV peptides. And as the reviewer rightly states, we did use all peptide identifications for statical evaluations of false discovery rates and could show that DIA has a great potential for this challenging task and might be even better suited than DDA. But we agree that more data would be required for a comprehensive and final comparison. We have therefore revised the corresponding part in the conclusion and now write:
“The overall number of SAAV peptides identified by DIA-MS search engine pipelines ranged from 6 to 9 and was thus slightly higher than DDA (3-6 peptides). While the higher number of SAAV peptides identified by DIA is in line with the overall higher peptide numbers obtained, additional studies with a higher number of SAAV peptide identifications will be needed for a thorough comparison of the two MS workflows. Notably, the numbers of identified SAAV peptides appears relatively low compared to the 233 HeLa extracted variants. “